# Detecting alternative attractors in ecosystem dynamics

Torbjörn Säterberg [1✉] & Kevin McCann[2]

Dynamical systems theory suggests that ecosystems may exhibit alternative dynamical attractors. Such alternative attractors, as for example equilibria and cycles, have been found in the dynamics of experimental systems. Yet, for natural systems, where multiple biotic and abiotic factors simultaneously affect population dynamics, it is more challenging to distinguish alternative dynamical behaviors. Although recent research exemplifies that some natural systems can exhibit alternative states, a robust methodology for testing whether these constitute distinct dynamical attractors is currently lacking. Here, using attractor reconstruction techniques we develop such a test. Applications of the methodology to simulated, experimental and natural time series data, reveal that alternative dynamical behaviors are hard to distinguish if population dynamics are governed by purely stochastic processes. However, if population dynamics are brought about also by mechanisms internal to the system, alternative attractors can readily be detected. Since many natural populations display evidence of such internally driven dynamics, our approach offers a method for empirically testing whether ecosystems exhibit alternative dynamical attractors.

[1] Swedish University of Agricultural Sciences, Department of Aquatic Resources, Öregrund, Sweden. [2] Department of Integrative Biology, University of Guelp, Guelph, ON, Canada. ✉email: torbjorn.saterberg@slu.se

D ynamical systems theory has been applied extensively in the quest for a better understanding of critical transitions; that is, abrupt changes in the dynamics, of ecosystems[1–4]. A key result derived from this theory, indeed a common feature of non-linear mathematical models, is that novel dynamical regimes may arise through perturbations affecting either parameter values (that is, through bifurcations) or state variables of a system (systems can enter alternative basins of attraction)[2]. Even a simple discrete one-dimensional model can display a wide array of different types of dynamics[5], and in higher dimensions - the dimensionality of which real ecosystems most likely are composed - the set of potential dynamics that can arise from a model can increase dramatically[6]. Thus, if non-linear mathematical models are useful descriptions of real ecosystems we would expect some ecosystems to display qualitative changes in their temporal dynamics following critical transitions.

Experiments clearly indicate that alternative dynamical regimes, including alternative attractors such as cycles and chaos, can exist in a system[7–10]. Still, a robust proof of whether a given ecosystem has switched to an alternative attractor is lacking[3]. A number of studies indicate that natural systems may exhibit alternative attractors[3,11–15], yet to our knowledge, only one study has thoroughly investigated if the temporal dynamics in an ecosystem is qualitatively different pre and post a critical transition. In this study, it was found through visual inspection of time series that trajectories were qualitatively different pre and post an induced trophic cascade in a whole lake experiment[16]. However, no formal statistical test was conducted to determine whether the time series trajectories were indeed different. In fact, no such test has so far been developed.

Here we develop a formal test for testing whether systems exhibit alternative dynamical attractors. The test distinguishes if the trajectories of two-time series (e.g., pre-chosen from break-point analysis[17] or biological knowledge of the system) are qualitatively different. It is based on attractor reconstruction techniques (See "Methods") and the rationale behind the test is that predictions of the dynamics in a given dynamical regime should be significantly more accurate if time series from the same, rather than a contrasting dynamical regime, are used to inform predictions. Thus, if regimes (attractors) are dynamically dissimilar we expect significantly lower prediction errors for within than across regime (attractor) predictions, and therefore test if prediction errors of within and across regime predictions are significantly different.

## Results and discussion

We illustrate the approach using a simple example with two dynamical regimes (Fig. 1): a two-point limit cycle and a four-point limit cycle ($M_A$ and $M_B$ in Fig. 1, respectively). The trajectories for two species, a consumer $C$ and its resource $R$, can either be plotted as a function of time (left panels Fig. 1a) or in phase space (right panels Fig. 1a). Now, assume that information from one dynamical regime (e.g. $M_A$) is used to predict dynamics from the same dynamical regime, then predictions ($\hat{Y}_A(t)|M_A$) are very similar to the true observed dynamics (comparing $\hat{Y}_A(t)|M_A$ and $Y_A(t)$ in Fig. 1b). However, if information from another dynamical regime ($M_B$) is used to inform predictions, then predictions ($\hat{Y}_A(t)|M_B$) are completely different to the true dynamics (comparing $\hat{Y}_A(t)|M_B$ and $Y_A(t)$ in Fig. 1b). Predictions based on data from a contrasting regime ($\hat{Y}_A(t)|M_B$) are thus less accurate than predictions based on data from the same regime ($\hat{Y}_A(t)|M_A$) (Fig. 1c), inferring larger prediction errors for across regime predictions ($\hat{Y}_A(t)|M_B - Y_A(t)$) than within regime predictions ($\hat{Y}_A(t)|M_A - Y_A(t)$) (Fig. 1d). Moreover, larger across

($\hat{Y}_B(t)|M_A - Y_B(t)$) than within regime prediction errors ($\hat{Y}_B(t)|M_B - Y_B(t)$), would also be found if dynamical regime $Y_B(t)$ was predicted, as within regime predictions ($\hat{Y}_B(t)|M_B$) are more similar to the true dynamics than across regime predictions ($\hat{Y}_B(t)|M_A$) also for this dynamical regime ($Y_B(t)$) (last row Fig. 1b). A comparison of prediction errors for across and within regime predictions can thus be used to test if the temporal dynamics in two dynamical regimes are qualitatively different.

Figure 2 shows the probability of detecting difference in across ($\hat{Y}_i(t)|M_j - Y_i(t)$) and within regime ($\hat{Y}_i(t)|M_i - Y_i(t)$) prediction errors, in data from a simulated food-chain model (See "Methods" and Supplementary Figs. 1a–d and 2, 3). Differences in across and within regime prediction errors is detectable across all dynamical regimes in which the observed dynamics is a cyclic regime (i.e. B, C, or D in Fig. 2), except for the obvious case where across and within regime predictions are based on the same dynamical regime (diagonal in Fig. 2). For the specific case where across and within regime predictions are compared across a stochastic equilibrium and a cyclic regime (A predicts B, C, or D; and B, C, or D predicts A in Fig. 2), a dichotomous response is observed. This means that for a given pair of regimes, one test indicates a significant difference in prediction errors (e.g. A predicts B in Fig. 2) whereas in the opposite direction (e.g. B predicts A in Fig. 2) the test is not significant. This result is caused by the low predictive ability, and the low intrinsic predictability, of stochastic equilibria (Supplementary Table 1). Overall, our approach can thus robustly detect dynamical difference among contrasting regimes when at least one regime is governed by internally driven signals.

Fussman et al.[7] proposed that two qualitatively different dynamical regimes were apparent in an experimental system. This conclusion was arrived at through a comparison of the predictions of a mathematical model and the coefficient of variance of time series produced by experimentally varying the dilution rate in predator-prey chemostats. Two distinct dilution rate regions, giving rise to equilibria and cycles, were suggested based on an abrupt increase in the variability of time series at a specific dilution rate. This suggests that the system had gone through a Hopf-bifurcation, yet we note that there was no formal test conducted on whether the temporal dynamics were qualitatively different in the different parameter regions.

Our analyses of the Fussman et al.[7] data set show that the temporal dynamics of time series (Supplementary Figs. 4–11), previously suggested to represent equilibria and cycles, are indeed different (Fig. 3). This is because: (i) a significant difference in prediction error for across and within regime predictions is often found when an "equilibrium" time series is used to predict a "cyclic" time series (upper left squares Fig. 3a, b; note that the most obvious exceptions are observed for the two shortest time series [$n = 18$ and $21$]); and (ii) a difference in prediction errors for across and within regime predictions is not found when "cyclic" time series are used to predict "equilibria" (lower right squares Fig. 3a, b). These results indicate that the "equilibrium" time series are mainly stochastic, and that the "cyclic" time series contain a higher degree of predictable information; a result which resembles the result derived from analyses of data from the food-chain model (B predicts A, and A predicts B in Fig. 2).

Moreover, a significant difference in prediction errors for across and within regime predictions is often also found when "cyclic" time series are used to predict other "cyclic" time series (upper right squares Fig. 3a, b). This result may seem counter-intuitive. Yet, if a non-linear mathematical model is a good description of an experimental system, generically different cycle periodicities and amplitudes will emerge for different parameter values within the cyclic parameter region[6]. That we find

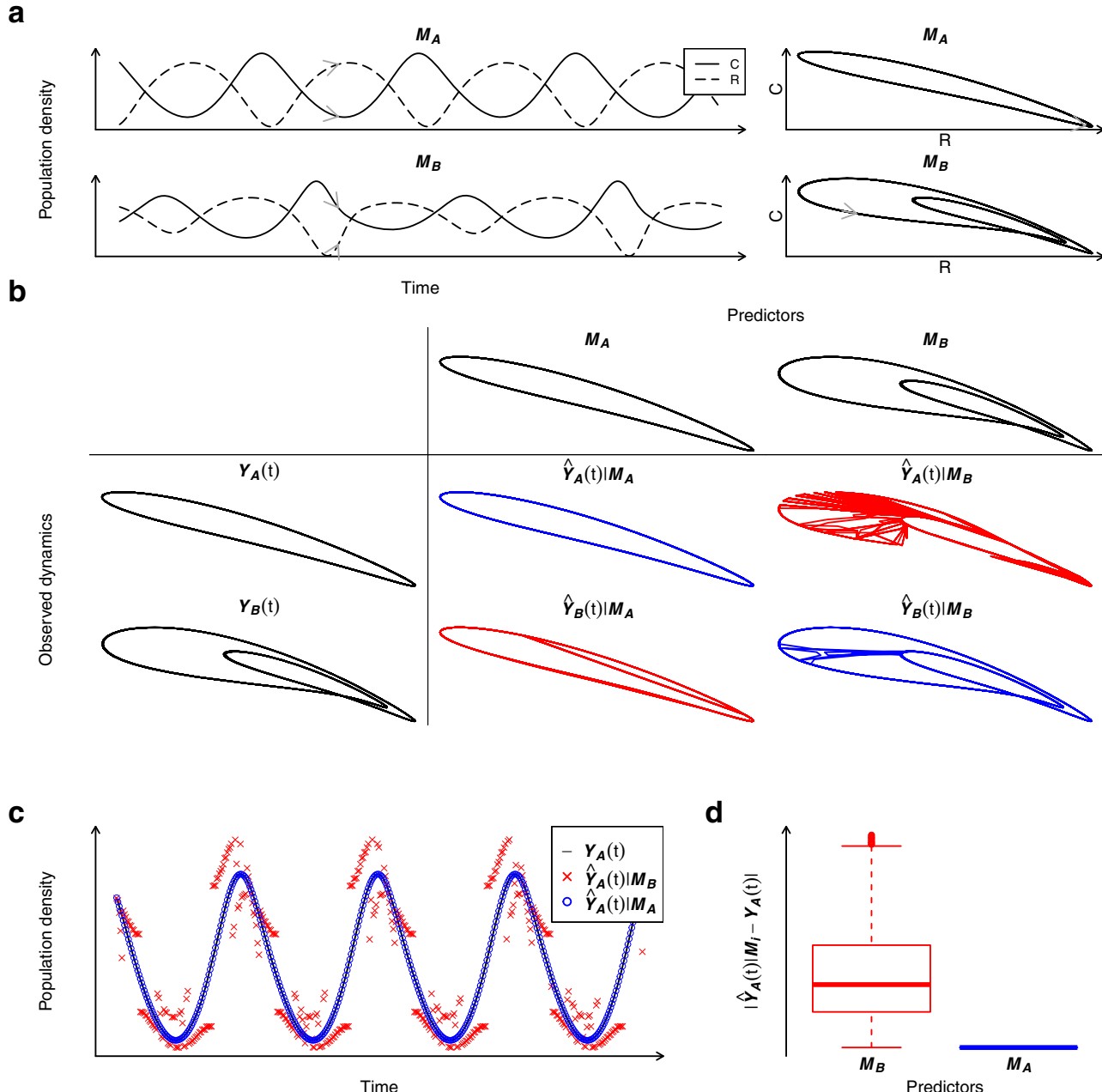

**Fig. 1 An illustration of the methodological approach of detecting alternative attractors in ecosystem dynamics. a** Time-series plot (left panels) and phase space plot (right panels) showing the dynamics of a consumer (*C*) and its resource (*R*) in a food-chain model. $M_A$ and $M_B$ are abbreviations for two dynamical regimes: a 2-point limit cycle and a 4-point limit cycle, respectively. The gray arrow illustrates the flow in phase space. **b** The observed dynamics of two different dynamical regimes ($Y_A$(t) & $Y_B$(t)) is estimated using information from either of two dynamical regimes ($M_A$ and $M_B$) giving rise to within ($\hat{Y}_A(t)|M_A$ and $\hat{Y}_B(t)|M_B$ [in blue]) and across regime predictions ($\hat{Y}_A(t)|M_B$ and $\hat{Y}_B(t)|M_A$ [in red]). **c** Within ($\hat{Y}_A(t)|M_A$) and across regime predictions ($\hat{Y}_A(t)|M_B$) for one species in one dynamical regime ($Y_A$(t)). **d** Absolute predictions errors for across and within regime predictions are used to test if the temporal dynamics within two contrasting regimes are dissimilar. Predator dynamics is here used to predict consumer and resource dynamics (See "Methods").

dynamical difference among "cyclic" time series may thus be explained by the fact that these time series indeed have qualitatively different internal signatures. Our results thus extend the findings of Fussman et al.[7] by showing that the temporal dynamics of unique time series, from a region of parameter values producing cyclic dynamics, are often also qualitatively different.

The experimental system and the food-chain model (Figs. 2, 3) investigated above are examples of deterministic systems, since mechanisms internal to those systems induce population variability. However, alternative dynamical regimes may also exist in systems where population dynamics is brought about by stochasticity. It is for example well-known that the characteristics of time series (e.g. autocorrelation) produced by stochastic one-dimensional models depend on their proximity to bifurcation points[4]. If a model exhibits alternative stable equilibria this infers that random perturbations may induce qualitatively different time-series signals depending on in which basin of attraction the model's state resides (Fig. 4a and Supplementary Fig. 1e–g).

We apply our approach to data produced by a stochastic alternative stable state model ("Methods"; Fig. 4a and

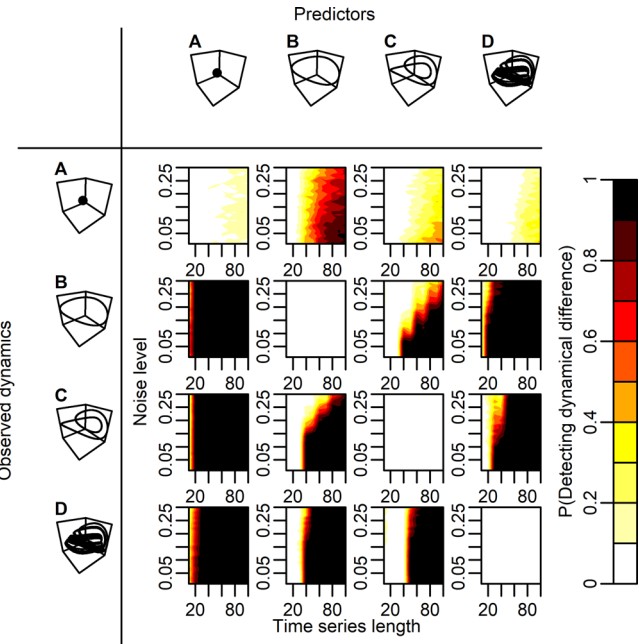

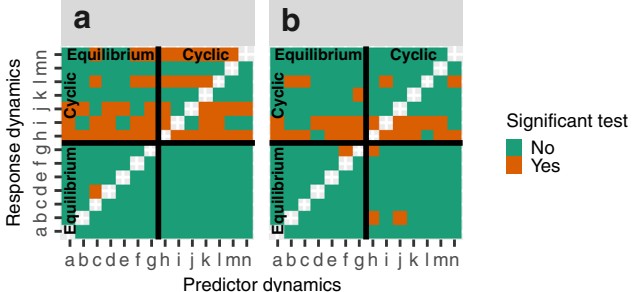

**Fig. 3 Detecting alternative dynamical attractors in an experimental data set[7].** This figure shows if there is significant difference in across ($|\hat{Y}_i|M_j(t) - Y_i(t)|$) and within regime prediction errors ($|\hat{Y}_i|M_i(t) - Y_i(t)|$) across each combination of 14 experimental time series (a($n = 87$), b ($n = 63$), c($n = 75$), d($n = 77$), e($n = 16$), f($n = 94$), g($n = 90$), h($n = 114$), i ($n = 120$), j($n = 41$), k($n = 18$), l($n = 49$), m($n = 21$) and n($n = 47$); Supplementary Figs. 4–11). These time series were earlier classified as either equilibria or cycles[7]; a classification which is here displayed by black thick lines. Significant, and non-significant, tests are illustrated in orange, and green, respectively (H0:$|\hat{Y}_i|M_i - Y_i| > |\hat{Y}_i|M_j - Y_i|$; permutation test; $p = 0.05$). Indexes ($i, j$) refer to row $i$ and column $j$ in the grids. Subpanels show cases where (**a**) *Brachionus calyciflorus*, and (**b**) *Chlorella vulgaris*, time series are used to predict *Chlorella vulgaris*, and *Brachionus calyciflorus*, time series, respectively (See "Methods").

**Fig. 2 Detecting alternative dynamical attractors in systems dominated by internally driven signals.** This figure shows the probability of detecting significant difference (P(Detecting dynamical difference); color bar) in absolute prediction error for across ($|\hat{Y}_i(t)|M_j - Y_i(t)|$) and within regime predictions ($|\hat{Y}_i(t)|M_i - Y_i(t)|$) in data produced by a food-chain model. The observed dynamical regimes, $Y_i(t)$, which are predicted using within ($\hat{Y}_i(t)|M_i$) and across regime dynamics ($\hat{Y}_i(t)|M_j$) are shown in the first column: A, equilibrium; B, a 2-point limit cycle; C, a 4-point limit cycle; D, a chaotic attractor. Across regime predictors, $M_j$, are displayed in the top row (See "Methods"). Time-series length and observation noise level are varied for each combination of predictor and response regime. Probabilities of detecting dynamical difference (the color bar) were derived by testing the null-hypothesis (H0:$|\hat{Y}_i|M_i - Y_i| > |\hat{Y}_i|M_j - Y_i|$; permutation test; $p = 0.05$) across 100 replicates for each combination of time series length and observation noise level. Time-series length was varied from 10 to 100 in steps of 10, and observation noise, $\rho$, was varied from 0.01 to 0.3 in steps of 0.01, in total yielding 300 combinations of observation noise and time series length, for each combination of dynamical regimes $i$ and $j$. In this example, predator dynamics is used to predict consumer and resource dynamics using the multivariate approach (See "Methods"; results for the cases where consumer or resource dynamics are used to predict the other species´ dynamics are presented in Supplementary Figs. 2, 3). All time series were standardized ($\mu = 0; sd = 1$) prior testing for dynamical difference.

Supplementary Fig. 1e–g). In general, alternative stochastic regimes are not distinguishable (Fig. 4b, c). Still, for fixed parameters values that may best represent natural systems that have gone through regime shifts (i.e. parameter values close to bifurcation points) dynamical difference is detectable ($c \approx 1.8$ or $c \approx 2.7$, in Fig. 4b, c, respectively). However, long time series are required for a robust detection also for these specific cases. Thus, our approach has a limited ability of detecting dynamical difference among alternative stochastic regimes.

As a final example of the approach, we apply it to a phytoplankton time series from a eutrophic Lake in Germany. We tested if pre- and post-critical transition time series, as previously found using breakpoint analysis[18], constitute alternative dynamical attractors in this system (Fig. 5a). The results show a significant difference ($p \approx 0.03$, permutation test) in across and within regime predictions for the pre-transition time series, and no difference ($p \approx 0.98$, permutation test) in across and within

regime prediction errors for the post-transition time series (Fig. 5b). This suggests that the pre-transition dynamics constitute a more strongly internally driven dynamical regime than the post-transition dynamics, a result resembling the result found when comparing cyclic and stochastic equilibria produced by a food-chain model (Fig. 2). A comparison of mean absolute prediction errors (MAPE) of pre- and post-transition dynamics further supports this assertion by showing that the post-transition dynamics are associated with overall larger prediction errors (MAPE$_W$ = 0.6; MAPE$_A$ = 0.67) than the pre-transition dynamics (MAPE$_W$ = 0.53; MAPE$_A$ = 0.57). Predictions for post-transition dynamics are thus less accurate than predictions of pre-transition dynamics, suggesting that the pre-transition dynamics is to a larger extent governed by internally driven dynamics than the post-transition dynamics.

## Conclusion

Overall, simulation results show that if internally driven signals are evident in one out of two, or both regimes, of a natural population time series, our methodology can robustly detect alternative dynamical attractors (See also Supplementary Discussion). However, if stochastic processes dominate in both regimes of a time series, the approach cannot distinguish alternative attractors. Therefore, if dynamical difference is detected using the methodology developed here alternative internally driven dynamics is detected. On the other hand, if a system is strongly driven by stochastic processes, alternative stochastic behaviors may be detected using other approaches such as power spectrum analyses[19].

Since the methodological approach presented in this study is most likely to detect dynamic dissimilarity among time series displaying internally driven signals, a key question is whether natural populations exhibit deterministic signals. To this end, meta-analyses of population time series have often found that natural populations display non-linear deterministic dynamics such as cycles[20,21], suggesting that internally driven signals are not overwhelmed by stochasticity in natural populations. In natural systems, stochastic processes most likely integrate with

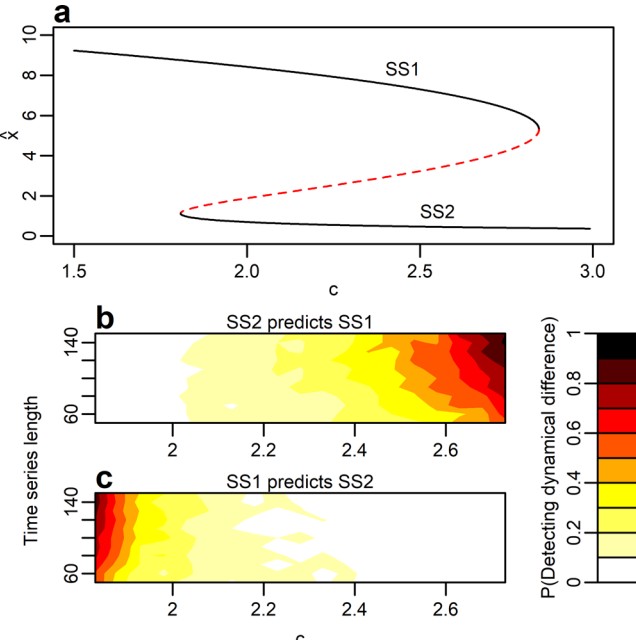

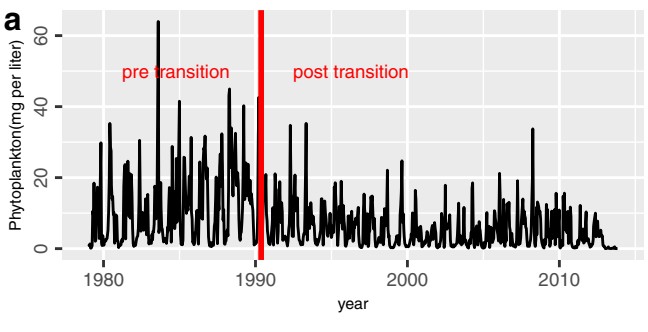

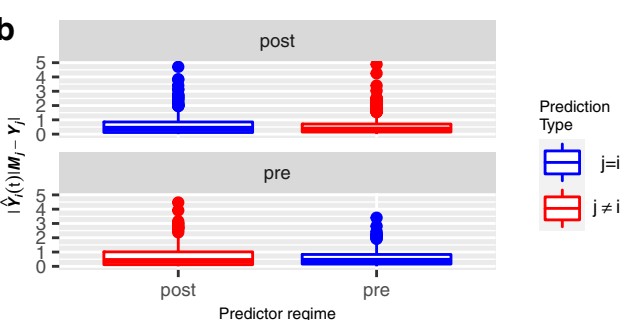

**Fig. 4 Detecting alternative attractors in alternative stochastic regimes.** This figure shows the probability of detecting significant difference ($P$ (Detecting dynamical difference); color bar) in absolute prediction errors for across ($|\hat{\boldsymbol{Y}}_i(t)|\boldsymbol{M}_j - \boldsymbol{Y}_i(t)|$) and within regime predictions ($|\hat{\boldsymbol{Y}}_i(t)|\boldsymbol{M}_i - \boldsymbol{Y}_i(t)|$) in data produced by a stochastic alternative stable state model (See "Methods"). **a** Model equilibria (stable [black solid lines] and unstable [red dashed line]) as a function of harvest rate, $c$, for the model skeleton. **b** The probability of detecting significant difference in absolute prediction error when SS2 is used to predict SS1 (i.e. across regime prediction). **c** As in (**b**) but here SS1 is used to predict SS2. Probabilities were derived by testing the null-hypothesis H0:$|\hat{\boldsymbol{Y}}_i|\boldsymbol{M}_i - \boldsymbol{Y}_i| > |\hat{\boldsymbol{Y}}_i|\boldsymbol{M}_j - \boldsymbol{Y}_i|$ (permutation test $p = 0.05$) across 100 simulated data sets for each combination of time series length and harvest rate, $c$. Time-series length was varied between 50 and 150 in steps of 10, and harvest rate $c$ was varied between 1.83 and 2.73 in steps of 0.05, in total yielding 209 combinations of time series length and harvest rate. Each time series was standardized ($\mu = 0; sd = 1$) prior testing for difference in temporal dynamics of contrasting regimes.

internal processes[22,23], creating novel dynamics that necessarily still reflects the internal process[22–24]. As our approach is useful for such dynamics, we reckon it as part of the growing toolbox of approaches (e.g. early warning signals) required to understand and mitigate against ecosystem collapse (e.g. regime shifts). With the increasing presence of the global footprint development of such methodologies are critical.

## Methods

**Detecting alternative attractors in ecosystem dynamics.** We use empirical dynamical modeling, a set of equation-free tools for analyzing non-linear time series (for a review and assumptions see[25,26], respectively), to test if the temporal dynamics of alternative dynamical regimes are qualitatively different. Empirical dynamic modeling builds fundamentally on Takens embedding theorem, which shows that attractors of multi-dimensional dynamical systems can be reconstructed using higher order lags of its embedded time series[27]. However, if a dynamical system has gone through a bifurcation, or switched to an alternative basin of attraction, attractors are qualitative dissimilar in the two regimes. Theoretically, this infers that it should be possible to reconstruct the attractor of one regime using information from the same regime, but not from the other regime. In practice, this implies that if a model (attractor reconstruction) based on one dynamical regime is used to predict the dynamics of variables from the same dynamical regime pre-dictions should be accurate (i.e. low prediction errors), whereas if an attractor reconstruction based on one dynamical regime is used to predict the dynamics of variables of another attractor predictions should be less accurate (i.e. high pre-diction errors). We make use of this idea by specifically testing if prediction errors

**Fig. 5 Detecting alternative attractors in a phytoplankton time series from Lake Müggelse[18]. a** A phytoplankton time series (mg L$^{-1}$) from Lake Müggelse (black line). The red vertical line shows a breakpoint[18] separating two potential dynamical regimes: pre- and post-critical transition dynamics, which are here used to test for alternative dynamical attractors. **b** The upper panel shows prediction errors for the case where post-transition dynamics is predicted using either across regime dynamics ($|\hat{\boldsymbol{Y}}_i|\boldsymbol{M}_j - \boldsymbol{Y}_i| \,\epsilon\, j \neq i$; in red), that is pre-transition dynamics, or within regime dynamics ($|\hat{\boldsymbol{Y}}_i|\boldsymbol{M}_j - \boldsymbol{Y}_i| \,\epsilon\, j = i$; in blue), that is, post-transition dynamics. The lower panel shows the case where pre-transition dynamics is predicted using either across regime dynamics ($|\hat{\boldsymbol{Y}}_i|\boldsymbol{M}_j - \boldsymbol{Y}_i| \,\epsilon\, j \neq i$; in red), that is, post-transition dynamics, or within regime dynamics ($|\hat{\boldsymbol{Y}}_i|\boldsymbol{M}_j - \boldsymbol{Y}_i| \,\epsilon\, j = i$; in blue), that is, pre-transition dynamics. Mean absolute prediction errors for across and within regime predictions are significantly different for the pre-transition dynamics (H0:$|\hat{\boldsymbol{Y}}_i|\boldsymbol{M}_i - \boldsymbol{Y}_i| > |\hat{\boldsymbol{Y}}_i|\boldsymbol{M}_j - \boldsymbol{Y}_i|$; $P\approx 0.03$; permutation test), but not for post-transition dynamics (H0:$|\hat{\boldsymbol{Y}}_i|\boldsymbol{M}_i - \boldsymbol{Y}_i| > |\hat{\boldsymbol{Y}}_i|\boldsymbol{M}_j - \boldsymbol{Y}_i|$; $P\approx 0.98$; permutation test). Mean absolute prediction errors for within (MAPE$_W$) and across regime predictions (MAPE$_A$), for the post- and pre-transition dynamics, are given by 0.6 ($n = 610$), 0.53 ($n = 607$), 0.57 ($n = 291$) and 0.67 ($n = 294$), respectively. The univariate prediction algorithm is here used to do predictions (See "Methods"). Boxplots show the median (center line), upper and lower quartiles (box limits), 1.5 x interquartile range (whiskers) and outliers (points). Each part of the time series was standardized ($\mu = 0; sd = 1$) prior testing for difference in temporal dynamics of contrasting regimes.

of across and within regime predictions are different. As explained below this idea can be used for both univariate and multivariate time series data.

**Univariate approach.** Univariate attractor reconstructions can be found using the simplex algorithm[28,29]. First, for a given dynamical regime, a time series can be split into a library of vectors, and each vector is described by

$$\underline{y}_A(t) = <Y_A(t), Y_A(t-1), Y_A(t-2), \dots, Y_A(t-(E-1))>, \qquad (1)$$

where $Y_A(t)$ is an observation of variable $Y$ at time $t$ in dynamical regime $A$ and $E$ is the reconstructed attractors embedding dimension. Using the simplex projection algorithm, a one-step ahead forecast is produced as follows:

$$\hat{Y}_A(t+1)|\boldsymbol{M}_B = \sum_{m=1\dots E+1} w_m Y_B(t_m + 1), \qquad (2)$$

where $t_m$ is a time index of an observation in dynamical regime $B$, $E$ is the embedding dimension of regime $B$, and $w_m$ is an exponential weighting

described by:

$$w_m = u_m \Big/ \sum_{n=1,\ldots,E+1} u_n, \tag{3}$$

where n and m belongs to the set of the E+1 nearest neighbors of vector $\underline{y}_A(t)$ in the set of vectors $\{\underline{y}_B(t_m)\}$, $u_m = \exp\{-d[\underline{y}_A(t), \underline{y}_B(t_m)]/d[\underline{y}_A(t), \underline{y}_B(t_1)]\}$, and $d[\underline{y}_A(t), \underline{y}_B(t_1)]$ is the Euclidean distance between the prediction vector $\underline{y}_A(t)$ and its nearest neighbor $\underline{y}_B(t_1)$ in the set $\{\underline{y}_B(t_m)\}$.

The only parameter that is estimated using the simplex algorithm is the embedding dimension E. This parameter is estimated by optimizing the correlation between observations ($Y_A(t+1)$) and predictions ($\hat{Y}_A(t+1)|M_A$) using a leave-one-out cross validation approach (See Supplementary Discussion). The embedding dimension E and its corresponding set of E-dimensional vectors (Eq. 1) constitutes the reconstructed attractor, $M_A$, of a given dynamical regime A. This reconstructed attractor ($M_A$) is then used to predict data for both the same dynamical regime ($\hat{Y}_A(t+1)|M_A$), and the contrasting dynamical regime $\hat{Y}_B(t+1)|M_A$. Likewise, the reconstructed attractor $M_B$ can be used to predict time series dynamics from both dynamical regimes; that is, $\hat{Y}_A(t+1)|M_B$ and $\hat{Y}_B(t+1)|M_B$, respectively.

**Multivariate approach**. A multivariate time series describes a number of simultaneously evolving variables. For example, a bivariate time series can be described by variables X and Y. For such time series, Sugihara et al.[30] developed an approach for testing if two variables (time series) are dynamically coupled. Their methodology builds on the fact that a reconstructed attractor should map 1:1 to the original attractor on which the reconstruction is based. This infers that two attractor reconstructions (based on two different variables) should also map 1:1 to each other[30]. Practically, this means that if two variables are dynamically coupled one-time series should be predictable based on an attractor reconstruction of another variable. However, if a dynamical system has gone through a bifurcation, or potentially switched to an alternative basin of attraction, a new set of rules will govern the dynamics of the system. Hence, a new attractor should have emerged. Now, since this new attractor is most likely governed by a new set of rules it should be difficult to predict the dynamics of this new alternative attractor based on information from the former attractor. Thus, if one variable in one dynamical regime is used to predict another variable in another dynamical regime, predictions should be biased. Yet, if one variable from one dynamical regime is used to predict another variable from the same regime predictions should be more accurate.

The simplex algorithm can be used to make predictions of a variable Y using a time series of another variable X[30]. Predictions are produced as follows:

$$\hat{Y}_A(t)|M_B = \sum_{m=1\ldots E+1} w_m Y_B(t_m), \tag{4}$$

where $t_m$ is the time series index of a vector of variable X of dynamical regime B, $w_m$ is an exponential weighting based on variable X:

$$w_m = u_m \Big/ \sum_{n=1,\ldots,E+1} u_n, \tag{5}$$

where n and m belongs to the set of the E+1 nearest neighbors of $\underline{x}_A(t)$ in $\{\underline{x}_B(t_m)\}$, $u_m = \exp\{-d[\underline{x}_A(t), \underline{x}_B(t_m)]/d[\underline{x}_A(t), \underline{x}_B(t_1)]\}$, and $d[\underline{x}_A(t), \underline{x}_B(t_1)]$ is the Euclidean distance between the prediction vector $\underline{x}_A(t)$ and its nearest neighbor $\underline{x}_B(t_1)$ in dynamical regime B.

The reconstructed attractors, $M_A$ and $M_B$, for each variable and regime are found using the univariate simplex algorithm described above[28–30]. Similar to the univariate case, the reconstructed attractor ($M_A$) is used to predict data from the same dynamical regime ($\hat{Y}_A(t)|M_A$), and to predict time series of a contrasting dynamical regime ($\hat{Y}_A(t)|M_B$). Yet, it is important to stress that $M_A$ here reflects an attractor reconstruction based on a variable that is not being predicted (that is, variable X is used to predict variable Y). This prediction approach thus infers that predictions are made on data that was not used to fit the model (X predicts Y and vice versa). Thus, neither across nor within regime predictions are made on data used to fit a model.

**Test statistic**. We used mean absolute prediction errors to test for difference between across and within regime predictions. Alternative metrics, such as mean sum of square errors, can also be used. However, since our approach gives skewed prediction errors we used mean absolute prediction errors to reduce the impact of extreme values. Further, since the absolute prediction errors are non-normally distributed we used a permutation test. The null hypothesis that is tested reads:

$$H0 : \mathrm{MAPE}_A < \mathrm{MAPE}_w, \tag{6}$$

where $\mathrm{MAPE}_A$ is the mean absolute prediction error for across regime predictions (that is, $\mathrm{MAPE}_A = \frac{1}{n}\sum_{t=1:n} \mathrm{abs}(\hat{Y}_{M_A}(t)|M_B - Y_{M_A}(t))$, and $\mathrm{MAPE}_w$ is the mean absolute prediction error for within regime predictions (that is, $\mathrm{MAPE}_w = \frac{1}{n}\sum_{t=1:n} \mathrm{abs}(\hat{Y}_{M_A}(t)|M_A - Y_{M_A}(t))$. A test is consider significant if observed difference in across and within regime mean prediction errors is larger than the 95th percentile of 1000 permuted data sets.

**Food-chain model**. We used a food-chain model parameterized as in McCann and Yodzis[31] to simulate food-chain dynamics:

$$\frac{dR}{dt} = R\left(1 - \frac{R}{K}\right) - \frac{x_c y_c CR}{R + R_0} \tag{7}$$

$$\frac{dC}{dt} = x_c C\left(-1 + \frac{y_C R}{R + R_0}\right) - \frac{x_P y_P PC}{C + C_0}$$

$$\frac{dP}{dt} = x_P P\left(-1 + \frac{y_P C}{C + C_0}\right),$$

where R is the resource density, C consumer density, and P predator density. All parameters, except half-saturation constants $R_0$ (here set to 0.16129) and $C_0$ (here set to 0.5), and resource carrying capacity K, are derived from bioenergetics and body size allometry[30] (xc = 0.4, yc = 2.009, yp = 2.876, R0 = 1, xp = 0.08). This model can display a rich set of dynamics depending on parameter values[31]. Here we alter resource carrying capacity K in order to simulate the dynamics (using the deSolve package[32] in R) of qualitatively different attractors (See Supplementary Fig. 1; K = 0.78, equilibrium; K = 0.85; two-point limit cycle; K = 0.92, four-point limit cycle; K = 0.997, chaotic dynamics). Every fifth time step of the simulated dynamics, corresponding to a sampling frequency of ≈10 samples per cycle for the 2-point limit cycle, was sampled. Observation noise was thereafter added to the deterministic dynamics produced by the model:

$$N_l(t) = N_l'(t) + \rho * e(t); e(t) \sim N(0, \sigma_{N'_l}), \tag{8}$$

where $N_l'(t)$ is the abundance of species l (P, C or R) simulated by the food-chain model at time point t, $\rho$ is the level of observation noise and $\sigma_{N'_l}$ is the standard deviation of the deterministic dynamics of species l produced by the food chain model.

In order to investigate how time series length and observation noise affects the probability of detecting alternative attractors we derived probability landscapes. These were derived by testing the null-hypothesis (H0:$|\hat{Y}_i|M_i - Y_i|>|\hat{Y}_i|M_j - Y_i|$; See Test statistic above) across 100 replicates for each combination of time series length and level of observation noise, $\rho$. Time-series length was varied from 10 to 100 in steps of 10, and observation noise, $\rho$, was varied from 0.01 to 0.3 in steps of 0.01, in total yielding 300 combinations of observation noise and time series length, for each combination of dynamical regimes i and j. Predator dynamics was used to predict consumer and resource dynamics using the multivariate approach described above (results for the cases where consumer or resource dynamics are used to predict the other species´ dynamics are presented in Supplementary Figs. 2, 3). All time series were standardized ($\mu = 0$; $sd = 1$) prior testing for dynamical difference.

**Experimental data set**. The experimental data set was given by Fussman et al.[7]. This data set contains 14 time series of a predator *Brachionus calyciflorus* and its prey *Chlorella vulgaris* derived from chemostat experiments. Time series for different dilution rates were produced by keeping the dilution rate fixed in different chemostats (Supplementary Figs. 3–11). *Brachionus calyciflorus* and *Chlorella vulgaris* time series were used to predict *Chlorella vulgaris* and *Brachionus calyciflorus* time series, respectively, using the multivariate approach described above. We tested for qualitative difference in the temporal dynamics across all time series, which were standardized ($\mu = 0$; $sd = 1$) prior testing.

**Alternative stable state model**. We used a stochastic version of a well-known alternative stable state model[4,33] to produce alternative stochastic dynamical regimes. The model is described by:

$$dx = \left(x\left(1 - \frac{x}{K}\right) + \frac{cx^2}{1 - x^2}\right)dt + \sigma dw, \tag{9}$$

where K is the carrying capacity (here set to 11), c is a harvest rate, and σ (here set to 0.01) is the magnitude of noise which is described by a Wiener process (dw).

The model was simulated for fixed harvest rates (c) assuming that the system state resides in either of its two basins of attraction. The initial value for the simulation was set to the equilibrium of the noise-free model skeleton for fixed harvest rates c, and σ is set low in order to avoid stochastic flips, so-called flickering, between alternative basins of attraction. Dynamics was integrated ($\Delta t = 0.01$) using the matlab-package SDE-Tools[34].

In order to investigate how time-series length and harvest rate, c, affects the probability of detecting alternative attractors in stochastic regimes we derived probability landscapes.

These were derived by testing the null-hypothesis H0:$|\hat{Y}_i|M_i - Y_i|>|\hat{Y}_i|M_j - Y_i|$ (permutation test p = 0.05) across 100 simulated data sets for each combination of time series length and harvest rate, c. Time-series length was varied between 50 and 150 in steps of 10, and c was varied between 1.83 and 2.73 in steps of 0.05, in total yielding 209 combinations of time series length and harvest rate. Each time series was standardized ($\mu = 0$; $sd = 1$) prior testing for difference in temporal dynamics of contrasting regimes.

**Natural time-series data**. In a previous study on early warning signals of impending regime shifts, Gsell et al.[18] used breakpoint analysis to identify two potential alternative dynamical regimes. We here test if these two-time series segments constitute alternative dynamical attractors. Prior analysis, we imputed a few missing observations (n = 24) using a kalman smoother[35]. The two time series segments, i.e. pre- and post-breakpoint time series, were standardized ($\mu = 0; sd = 1$) prior testing for dynamical difference.

**Reporting summary**. Further information on research design is available in the Nature Research Reporting Summary linked to this article.

## Data availability

The experimental data set and the phytoplankton time series analyzed in this study were gathered from two previous studies[7,18] and all other data was simulated using theoretical models. All figures have associated raw data. The phytoplankton time series is available through the IGB database[36] and all other data, including experimental data from Fussmann et al.[7], is available through an open repository[37].

## Code availability

The code and simulated data for conducting all analyses and reproducing the figures in this manuscript are publically available[37].

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

## Acknowledgements

The authors thank Kevin Cazelles and Os Schmitz for reviewing this manuscript, and Gabriel Gellner, David Gilljam and Anna Gårdmark for discussions. We are grateful to Gregor Fussmann who provided experimental data to this work. We acknowledge the Leibniz Institute of Freshwater Ecology and Inland Fisheries (IGB) for providing long-term data from Lake Müggelsee. We thank the Integrative Biology Department at the University of Guelph, Canada and the Institute of Coastal Research (SLU) in Öregrund, Sweden for inspiring research environments. This work was financed through the Swedish Research Council FORMAS (no. 2017-00433) to T.S., and food from Thought grant to K.M.

## Author contributions

T.S. conceived the idea, T.S. and K.M. developed the methodology, T.S. wrote computer code and created figures, T.S. wrote the initial draft, T.S and K.M. reviewed and edited the manuscript. T.S. acquired the financial support leading to this publication.

## Funding

## Competing interests

The authors declare no competing interests.
