## [Peer Review File · Communications Biology]

Reviewers' Comments:

Reviewer #1:

Remarks to the Author:

This MS presents an important new advance in testing for alternative dynamical regimes in ecological time series. The MS builds a logic starting with simple dynamical systems, moving to more complex multitrophic systems, thus covering the gamut of empirical systems for which time series data have been collected. The MS is written for a readership that has advanced, sophisticated understanding of mathematics, which is fine for this targeted audience that perhaps does work to test for dynamical regimes in empirical time series. But the MS will not be accessible to empiricists who largely collect time series data in natural field studies—for which this new application is intended. I think the work could be made more broadly accessible, but the MS would need to be revised to expand the narrative to explain things more intuitively (even though it already does a reasonable job of doing this).

One place where there could be expanded narrative is in the paragraph comprised of line 50-55. The empirical issue here is that if you cannot identify alternative dynamical regimes a priori in a time series, then how do you go about in practice deciding a priori and objectively which part of the time series belongs to regime A vs regime B? I appreciate that this is explained, albeit at a high level of technical (and a bit abstract) detail, in the Methods when discussing using Takens theorem. I think more can be done in in paragraph line 50-55 to explain this better to the readership, as this is the important crux of all subsequent tests. Moreover, the section beginning at line 217 could be further elaborated to “walk” the reader through an example of how this would be done on a time series. I don't mean it to be heavily didactic, but it would help at least to give an illustrative example.

The symbolics in line 62, e.g., $\hat{Y}|MA(t)$ are incongruent with the corresponding form in Fig. 1 row 1 column 1, i.e. $\hat{Y}(t)|MA$. The same is true for the other predictor.

Lines 57-70 describe how testing for prediction error works using the first row and both columns of Fig. 1b—which makes things intuitive. But the MS would be strengthened if it explained also how the second row and the two columns work, because in this case both predictors seem to operate poorly. Why is this?

Figure 2. I don't understand what the hexagonal figures represent. They look like phase spaces but why use a hexagon rather than a square?

Figure 3, which addresses the empirical case study is extremely abstract. Given that the example comes from real world data, it would help to label the quantitative domains of the axes in the figure. Right now, there is no way to interpret what the color patterns mean and why they even stack contingently the way they do in 2-D space.

Further to this, most of the case study is addressed in extended figures, that are not part of the main MS. I assume they will be published on-line. This makes for a seriously annoying slog to go through each of the figures one by one (and would still be a slog if it is part of the published PDF). My advice is to develop this part as a stand-alone coherent case study gathered together (perhaps in a single downloadable pdf) that describes the application of Takens embedding theory and how predictions were made and tested statistically.

Line 135. How do you know a priori whether or not a time series is largely driven by internal dynamics or stochasticity. Given that this is a key issue concerning the reliability of the proposed test with an existing time series (i.e., only works when internally-driven dynamics dominate) more needs to be discussed on the issue, especially how one would a priori test for the dominance of one or the other driver. The MS largely punts on this issue in the closing section of the MS.

Os Schmitz

Reviewer #2:

Remarks to the Author:

In this article, the authors used attractor reconstruction techniques in order to develop a test of whether ecosystems exhibit alternative dynamical attractors. Although the idea is interesting, it is not new.

The article is purely speculative and does not contain any reference to a real-world deterministic ecosystem where such alternative dynamical attractors occur. The article needs to be reworked to meet the requirements of the journal. First, it is necessary to add an introduction section, and then the methodology. after that, this method should be applied to different ecosystems. Finally, the article should be included a conclusion section.

Reviewer #3:

None

We thank the reviewers for their constructive comments. See below for responses to reviewer comments (marked in yellow).

Reviewer #1 (Remarks to the Author):

1. This MS presents an important new advance in testing for alternative dynamical regimes in ecological time series. The MS builds a logic starting with simple dynamical systems, moving to more complex multitrophic systems, thus covering the gamut of empirical systems for which time series data have been collected. The MS is written for a readership that has advanced, sophisticated understanding of mathematics, which is fine for this targeted audience that perhaps does work to test for dynamical regimes in empirical time series. But the MS will not be accessible to empiricists who largely collect time series data in natural field studies—for which this new application is intended. I think the work could be made more broadly accessible, but the MS would need to be revised to expand the narrative to explain things more intuitively (even though it already does a reasonable job of doing this).

One place where there could be expanded narrative is in the paragraph comprised of line 50-55. The empirical issue here is that if you cannot identify alternative dynamical regimes a priori in a time series, then how do you go about in practice deciding a priori and objectively which part of the time series belongs to regime A vs regime B? I appreciate that this is explained, albeit at a high level of technical (and a bit abstract) detail, in the Methods when discussing using Takens theorem. I think more can be done in paragraph line 50-55 to explain this better to the readership, as this is the important crux of all subsequent tests. Moreover, the section beginning at line 217 could be further elaborated to “walk” the reader through an example of how this would be done on a time series. I don’t mean it to be heavily didactic, but it would help at least to give an illustrative example.

We have added clarifying text in the paragraph explaining the rationale behind the test (lines 51-53 & 57-58). Using the proposed methodology it is only possible to test if the trajectories of two pre-determined time series are qualitatively different. We hope that we have now clarified this issue.

2. The symbolics in line 62, e.g., $Y_{hat}|MA(t)$ are incongruent with the corresponding form in Fig. 1 row 1 column 1, i.e. $Y_{hat}(t)|MA$. The same is true for the other predictor.

Thanks for notifying us about this. It is now fixed!

3. Lines 57-70 describe how testing for prediction error works using the first row and both columns of Fig. 1b—which makes things intuitive. But the MS would be strengthened if it explained also how the second row and the two columns work, because in this case both predictors seem to operate poorly. Why is this?

We find the expected response also for this case. That is, within regime predictions are more accurate than across regime predictions for regime MB. We would argue that predictor MB inform very accurate predictions in this case (row 2 column 2), and that predictor MA gives less accurate predictions (row 1 column 2).

In order to not complicate figure 1 too much, we have chosen to keep it as it is, but have included one sentence discussing also this case (lines 72-76).

4. Figure 2. I don’t understand what the hexagonal figures represent. They look like phase spaces but why use a hexagon rather than a square?

We have added a more thorough explanation in the figure legend and hope that this will clarify what the figure represents (lines 406-507).

5. Figure 3, which addresses the empirical case study is extremely abstract. Given that the example comes from real world data, it would help to label the quantitative domains of the axes in the figure. Right now, there is no way to interpret what the color patterns mean and why they even stack contingently the way they do in 2-D space.

In order to illustrate that the grid represents statistical tests across each combination of 14 different time series we added letters to each grid of the figure. We hope that this clarifies that each square represents one statistical test.

6. Further to this, most of the case study is addressed in extended figures, that are not part of the main MS. I assume they will be published on-line. This makes for a seriously annoying slog to go through each of the figures one by one (and would still be a slog if it is part of the published PDF). My advice is to develop this part as a stand-alone coherent case study gathered together (perhaps in a single downloadable pdf) that describes the application of Takens embedding theory and how predictions were made and tested statistically.

We agree that having figures in many places is annoying. We have therefore added the field data example to the main text (lines 143-157 & 540-559 ; Fig. 5), and removed the results showing results from simulating a stochastic cyclic system. All other figures are just additional material, which complement the main findings.

7. Line 135. How do you know a priori whether or not a time series is largely driven by internal dynamics or stochasticity. Given that this is a key issue concerning the reliability of the proposed test with a existing time series (i.e., only works when internally-driven dynamics dominate) more needs to be discussed on the issue, especially how one would a priori test for the dominance of one or the other driver. The MS largely punts on this issue in the closing section of the MS.

In the paper we introduce a new approach for testing whether two time series constitute alternative dynamical attractors. The approach is applicable to both stochastic and more internally driven dynamics. However, the approach cannot distinguish two purely stochastic time series (as shown in figure 3) due to the lack of predictable information in such time series. Thus, it works best for distinguishing dynamics when at least one of the time series are associated with some type of cyclicity. We have added new text discussing this issue further (lines 159-172).

Os Schmitz

Reviewer #2 (Remarks to the Author):

8. In this article, the authors used attractor reconstruction techniques in order to develop a test of whether ecosystems exhibit alternative dynamical attractors. Although the idea is interesting, it is not new.

In this paper we develop a statistical test for testing whether time series from two regimes are qualitatively different, that is, if the two regimes constitute two different dynamical attractors. The method is thus amendable for testing if alternative attractors occur in a system. We are unaware of any formal statistical test that can be used to distinguish alternative dynamical attractors from ecosystem dynamics and therefore think that the approach is new.

9. The article is purely speculative and does not contain any reference to a real-world deterministic ecosystem where such alternative dynamical attractors occur. The article needs to be reworked to meet the requirements of the journal. First, it is necessary to add an introduction section, and then the methodology. After that, this method should be applied to different ecosystems. Finally, the article should be included a conclusion section.

We have referred to a number of real world studies suggesting that alternative dynamical states may occur in real ecosystem (e.g. refs 11-16), yet we do not know of any study that test if these constitute alternative dynamical attractors. Moreover, as we here introduce a novel approach of testing this issue we find it more appropriate to first robustly test the method against simulated data before applying it to real world data, and have therefore decided not to apply the approach to any more empirical examples. Therefore, we think that applying the approach to more empirical examples is out of the scope of the paper.

We have now formatted the manuscript as suggested.

Reviewers' Comments:

Reviewer #1:

Remarks to the Author:

The authors have addressed my comments and concerns to my satisfaction.